# Harnessing Phosphorous (P) Fertilizer-Insensitive Bacteria to Enhance Rhizosphere P Bioavailability in Legumes

**DOI:** 10.3390/microorganisms12020353

**Published:** 2024-02-08

**Authors:** Antisar Afkairin, Mary M. Dixon, Cassidy Buchanan, James A. Ippolito, Daniel K. Manter, Jessica G. Davis, Jorge M. Vivanco

**Affiliations:** 1Department of Horticulture and Landscape Architecture, Colorado State University, Fort Collins, CO 80523, USA; antisar.afkairin@colostate.edu (A.A.); mary.dixon@colostate.edu (M.M.D.); 2Department of Soil and Crop Sciences, Colorado State University, Fort Collins, CO 80523, USA; cassidy.buchanan@colostate.edu (C.B.); ippolito.38@osu.edu (J.A.I.); 3School of Environment and Natural Resources, The Ohio State University, Columbus, OH 43210, USA; 4Agricultural Research Service, United States Department of Agriculture, Fort Collins, CO 80526, USA; daniel.manter@usda.gov; 5Agricultural Experiment Station, Colorado State University, Fort Collins, CO 80523, USA

**Keywords:** legume, phosphorus bioavailability, phosphorus-insensitive, phosphorus, rhizosphere microbiome

## Abstract

Phosphorous (P) is widely used in agriculture; yet, P fertilizers are a nonrenewable resource. Thus, mechanisms to improve soil P bioavailability need to be found. Legumes are efficient in P acquisition and, therefore, could be used to develop new technologies to improve soil P bioavailability. Here, we studied different species and varieties of legumes and their rhizosphere microbiome responses to low-P stress. Some varieties of common beans, cowpeas, and peas displayed a similar biomass with and without P fertilization. The rhizosphere microbiome of those varieties grown without P was composed of unique microbes displaying different levels of P solubilization and mineralization. When those varieties were amended with P, some of the microbes involved in P solubilization and mineralization decreased in abundance, but other microbes were insensitive to P fertilization. The microbes that decreased in abundance upon P fertilization belonged to groups that are commonly used as biofertilizers such as *Pseudomonas* and *Azospirillum.* The microbes that were not affected by P fertilization constitute unique species involved in P mineralization such as *Arenimonas daejeonensis*, *Hyphomicrobium hollandicum, Paenibacillus oenotherae,* and *Microlunatus speluncae*. These P-insensitive microbes could be used to optimize P utilization and drive future sustainable agricultural practices to reduce human dependency on a nonrenewable resource.

## 1. Introduction

Phosphorus (P) is an essential nutrient for sustaining plant growth; yet, P fertilizers are made from a nonrenewable resource, rock phosphate. Rock phosphate mines are primarily found in China, the Middle East, Northern Africa, and the United States [1]. The US Geological Survey reported that the global phosphate rock supply stands at approximately 62 Gigatonnes, while the demand for P fertilizer is projected to grow by 2.5–3.0% annually [2]. At this rate, the world’s P reserves are estimated to last for approximately 125 years [2]. The highly reactive nature of phosphate anions causes them to be immobilized through precipitation with free Ca and Mg in alkaline soils and Fe and Al in acidic soils, thus making phosphate unavailable to plants across a wide pH range. In commercial agriculture, growers are encouraged to increase P fertilization to deal with P immobilization and to support crop yields [3,4,5,6]. The reservoir of immobilized P, known as legacy P, can be calculated as the difference between the inputs (mineral P fertilizer, atmospheric deposition, and weathering) and the outputs (P lost through surface runoff, subsurface flow, leaching, crop uptake, etc.) [7]. Legacy P stocks in soils have the potential to be extremely important in maintaining agricultural production once P fertilizer sources become scarce [8]. Thus, a current challenge faced by scientists is developing strategies to increase the bioavailability of the immobilized soil P phase.

One potential means by which immobilized P can become available is through interaction with certain microorganisms. For example, P solubilizing bacteria (PSB) can efficiently mineralize immobilized P through excretion of organic acids [9]. These organic acids enhance P solubilization through different mechanisms such as acidification, which involves a decrease in the rhizosphere pH [10]. In calcareous soils, PSB dissociate tricalcium phosphate [Ca_3_(PO_4_)_2_] bonds via polymeric substance formation, chelation, and ion-exchange mechanisms [10]. Phosphatase enzymes (e.g., phytase, phosphonatase, and CeP lyase) are another mechanism by which PSB release phosphate ions [11]. These enzymes break down high-molecular-weight organic phosphate into low-molecular-weight compounds, eventually releasing inorganic P to soils and, importantly, in the area where soil and roots interact—the rhizosphere. 

In the rhizosphere, an interplay occurs between roots and root exudates for the purpose of nutrient acquisition [12]. For example, leguminous crops such as chickpea (*Cicer arietinum*) and faba bean (*Vicia faba*) exude organic acids that increase P availability in the soil [13]. Research has also shown that root exudates change in response to soil P concentration levels. For instance, 3-hydroxypropionic acid and nicotinic acid significantly accumulated under P deficiency, and these compounds were able to solubilize phosphate under solid and liquid medium conditions [14]. 

Furthermore, roots can manipulate the soil microbial composition for the purpose of increasing nutrient availability. As a plant progresses through its lifecycle, the patterns of metabolite exudation undergo modifications, resulting in alterations in the rhizosphere microbiome structure and activity [15,16,17]. Under conditions of P deficiency, plants and microorganisms release more P-hydrolyzing enzymes, which help to mineralize organic legacy P to inorganic forms that are more accessible for uptake [18]. 

To better understand how plants can utilize soil legacy P, crops with increased nutrient acquisition abilities should be utilized as model systems. Members of Fabaceae, the legumes, are viable model crops for P studies because of their rapid growth habit, atmospheric N_2_ fixation, and genotypic variability in terms of P solubilization capabilities. For example, some legume genotypes have demonstrated advantages over others in enhancing productivity under P-deficient conditions [19,20]. Legume crops such as chickpea and pea (*Pisum sativum*) have enhanced productivity due to associations with endophytic microbes such as rhizobium and PSB [21]. In addition, white lupin (*Lupinus albus*), pigeon pea (*Cajanus cajan*), and chickpea release carboxylates into the rhizosphere [22,23]. Carboxylates can chelate metal cations, such as calcium (Ca^2+^), and thus release the P bound to those cations [23]. When compared to other crops in P-deficient conditions, common beans (*Phaseolus vulgaris*) showed increased yield, soil P availability, P uptake, and soil acid phosphatase activity [2]. 

Utilizing legumes as a model to understand the ability of these crops to utilize legacy P could enhance overall crop production while reducing future reliance on P fertilizer inputs. To begin to address potential future shortages in P fertilizers, the initial aim of this study was developed to identify legume varieties with the ability to highly associate with PSB. The subsequent aim was to identify unique microbes with the ability to solubilize and mineralize P in the presence or absence of P fertilizers. We hypothesized that specific legume varieties are uniquely equipped to cultivate P-fertilizer-insensitive PSB that enhance the utilization of legacy P, thus offering a sustainable solution to potential P fertilizer shortages and contributing to more efficient crop production systems. 

## 2. Materials and Methods

### 2.1. Plant and Soil Material, Fertilizer Characteristics, and Growth Conditions for Legumes

Three plant species were tested in the study: common beans, cowpeas (*Vigna unguiculata*), and peas. Six common bean varieties were tested representing different market classes: pinto (‘Cowboy’), Mayocoba (‘Claim Jumper’), light red kidney (‘Big Red’), black (‘Black Beard’), great northern (‘Andromeda’), and Anasazi. Common bean germplasm was retrieved from Northern Bean and Feed (Lucerne, CO, USA) (‘Cowboy’, ‘Claim Jumper’), Kelley Bean Company (Scottsbluff, NE, USA) (‘Big Red’, ‘Black Beard’, ‘Andromeda’), and Adobe Milling (Dove Creek, CO, USA) (Anasazi). Three varieties of cowpea (California #5, California #46, and Cp4906) were selected from the Colorado State University (CSU) collection. Two Austrian winter peas were used: ‘Seed Ranch’ (Odessa, FL, USA) and ‘Melrose’ from the University of Idaho (Moscow, ID, USA), respectively. Three dry pea varieties comprising two market classes were selected from ProGene (Othello, WA, USA): yellow (‘Koyote’, ‘Goldenwood’) and green (‘Vail’). 

Soil was collected from CSU’s Agricultural Research, Development, and Education Center (ARDEC; 40°36′36.9″ N, 104°59′38.2″ W). The upper 5 cm of soil was removed to avoid any surface accumulated P, and then soil was collected by hand from the 5–15 cm depth and placed in 19 L buckets. The soil, classified as a fine, smectitic, and mesic Aridic Argiustoll of the Nunn series [24], was transported to the laboratory and air-dried. Pots (110 × 110 × 130 mm) were filled with a mixture of one part sand (Quikrete Play Sand, GA, USA) and one part soil. The soil mix was characterized for pH, soluble salts, organic matter, nitrate-N, total P, Olsen P, and water holding capacity by Ward Laboratories (Kearney, NE; Table 1). The soil had low available P (Olsen P) but a high P legacy (total P). Triple superphosphate (0-46-0) was used in this study following the recommendations of CSU Extension [25], with 0.098 g TSP (0.045 g P_2_O_5_) (as finely ground triple superphosphate) applied to each pot (labeled as “fertilized”). Control pots/plants were not provided with P_2_O_5_ fertilizer (labeled as “unfertilized”). To establish consistent growth of the tested varieties, three seeds of each plant were uniformly dropped into the pots. After sprouting, pots were thinned to one plant per pot. Pots were watered daily to bring the soil moisture content to field capacity through the duration of the greenhouse trial.

Legume plants were grown in a greenhouse in the Plant Growth Facilities at CSU for approximately seven weeks under temperatures ranging from 18 to 21 °C during the day and 17 to 20 °C during the night. Relative humidity was ambient, appx. 70%. There was a 16 h photoperiod. Treatments were arranged in a randomized complete design, which included a total of 280 experimental units generated from 14 varieties, 2 fertilizer treatments, and 10 replications. Open access research randomizer software version 4.0 of was used to establish the design [26].

### 2.2. Rhizosphere Soil Collection and Plant Sampling

Rhizosphere soil samples were collected from each plant after seven weeks by excavating the roots and separating the surrounding bulk soil. Soil particles attached to the roots were carefully removed, and the remaining soil adhering to the roots was identified as rhizosphere soil. The collected rhizosphere soil was individually placed into 15 mL Falcon tubes and stored at −80 °C until DNA extraction. The fresh biomass of shoots was also collected and recorded immediately after harvest. The shoots were placed in a 70 °C oven for 4 days and then weighed.

### 2.3. Nutrient Analysis

The dried shoot samples were powder-ground. Then, the total P content in the plant shoot tissue was extracted via digestion with concentrated HNO_3_ and 30% H_2_O_2_, with the P concentration determined via inductively coupled plasma–atomic emission spectroscopy (ICP-OES). We utilized an ICP-OES instrument manufactured by PerkinElmer [27]. The plant-available P in the bulk soil samples was determined using Olsen-P extraction [28]. 

### 2.4. Soil Microbial DNA Extraction

Soil DNA extraction was performed on the 0.25 g rhizosphere samples, using a DNeasy Power soil PRO isolation kit and QIAcube (Qiagen, Hilden, Germany), according to the manufacturer’s instructions. The DNA was quantified using a Qubit Fluorometer (Invitrogen Qubit Flex Fluorometer).

### 2.5. 16S Amplicon Sequencing with Minion Flow Cells

To prepare the DNA for analysis, it was necessary to dilute the extracted DNA based on the Qubit concentrations (ng μL^−1^). Therefore, a dilution factor of 15 was applied using nuclease-free water. The mastermix used for PCR contained 20 μL Phusion HSII master mix, 14.4 μL water, 0.8 μL forward primer (10 µM), and 0.8 μL reverse primer (10 µM), resulting in a total of 36 μL of mastermix per 4 μL of sample. The bacterial primers used were Bact_27F-Mn (5′-TTTCTGTTGGTGCTGATATTGCAGRGTTYGATYMTGGCTCAG-3′) and Bact_1492R-Mn (5′-ACTTGCCTGTCGCTCTATCTTCTACCTTGTTACGACTT-3′) [29]. The PCR conditions were 98 °C for 30 s, followed by 25 cycles of 98 °C for 15 s, 50 °C for 15 s, and 72 °C for 60 s, and a final extension step of 72 °C for 5 min. The PCR products were purified using SPRI beads and 70% EtOH. The DNA was eluted in a 96-well plate with 40 µL PCR-grade water and quantified using a Qubit fluorometer. Using a second PCR reaction, DNA barcodes were attached to the amplicons to facilitate multiplexing. The second PCR conditions were 98 °C for 30 s, followed by 15 cycles of 98 °C for 15 s, 62 °C for 15 s, and 72 °C for 60 s, and a final extension step of 72 °C for 5 min.

Following the second PCR, the barcoded amplicons were purified using SPRI beads and 70% EtOH, and all samples were pooled into one multiplexed library. Sequencing was performed for 48 h at the USDA-ARS facility in Fort Collins, CO, using a MinION sequencer, R9.4.1 flow cell, and SQK-LSK109 sequencing kit. The raw data were processed using Guppy v6.0.1 to base-call and demultiplex the sequences, which were then filtered based on length (1000–2000 bp) and a minimum q-score of 70 using Filtlong v0.2.1. Vsearch was used to remove the chimeras and was employed to assign the taxonomy using the default NCBI-linked Reference Database from Emu. To correct the errors, Emu v3.0.0 was used with an expectation minimization algorithm that adjusts taxonomic assignments based on up to 50 sequence alignments per sequence read. Samples with less than 10,000 reads were excluded from downstream analyses.

To quantify the V3-V4 region of the 16S gene, the primer pair 341F/806R [30] was used for quantitative PCR in triplicate. The reaction mixture contained 10 µL Maxima SYBR-green (Thermo Scientific, Waltham, MA, USA), 2 µL of each 10 µM forward and reverse primer, 4 µL molecular grade H_2_O, and 2 µL soil DNA extract (diluted 1:20 with nuclease-free water) in a 20 µL reaction. The Roche 96 Lightcycler (Roche, Indianapolis, IN, USA) was used to run the reactions with an initial denaturation step at 95 °C for 5 min, followed by 28 cycles of denaturation at 95 °C for 40 s, annealing at 55 °C for 120 s, an extension at 72 °C for 60 s, and a final extension step at 72 °C for 7 min. To determine the amount of DNA, a standard curve was generated by serially diluting the purified *Pseudomonas putida* KT2440 gDNA, and the quantification cycle was compared. The copy numbers were then normalized based on the weight of the dry soil extract.

To estimate the abundance of functional genes classified by KEGG ontologies in the EMU reference database, we used PICRUSt2. This involved following the default PICRUSt2 pipeline, which had two steps. First, we used the python script place_seqs.py with EPA-NG to add the query sequences to the default PICRUSt2 prokaryotic 16S rRNA phylogenetic tree. Second, we used the python script hsp.py with the castor R package to predict the number of 16S rRNA and functional gene copies per genome [31].

### 2.6. Statistical Analysis

Data were analyzed using RStudio Version 2022.07.1. Significant differences between P treatments (i.e., fertilized versus unfertilized) and different plant varieties were analyzed using a two-way ANOVA to evaluate the significance of the differences in shoot biomass. Tukey adjusted pairwise comparisons were used to identify significant differences (*p* < 0.05) between treatments. A one-way ANOVA was used to evaluate significant differences in shoot P concentration, shoot P uptake, and available P in soil among different varieties [32]. Differential expression analysis was conducted on bacterial species with varying total abundances (measured in 16S copies per g of soil) at two different levels of P treatment, and an FDR p adjustment method was used to determine significant differences. The analysis was based on the negative binomial distribution and utilized the edge R package [33]. In addition, differential expression analysis was carried out on all operational taxonomic units (OTUs) that were mapped to the chosen P cycle genes. This analysis compared the total abundances between the unfertilized treatment and the fertilized treatment for all OTUs that shared 99% genetic distance. To further elucidate the differences in the whole bacterial community structure, a distance-based redundancy analysis (db-RDA) was run [31]. 

## 3. Results

### 3.1. Dry Shoot Biomass

The impact of P fertilizer on the shoot biomass was studied in three legume crop species and varieties. Common beans, including ‘Andromeda’, ‘Big Red’, and ‘Black Beard’, increased the dry shoot biomass when subjected to the P fertilizer treatment by 36%, 38%, and 45%, respectively. In contrast, no significant P fertilizer effect was observed in the other varieties (Anasazi, ‘Claim Jumper’, and ‘Cowboy’) (Figure 1A). 

For cowpeas, the shoot biomass of Cp4906 showed a substantial increase (54%) in shoot biomass in response to the P treatment when compared to the controls. However, for California #46 and California #5, no significant increase in biomass was observed upon P fertilization (Figure 1B). 

In peas, ‘Koyote’, ‘Melrose’, and ‘Seed Ranch’ showed increased dry shoot biomass (36%, 45%, and 33%, respectively) with the fertilized treatment compared to the unfertilized treatment (Figure 1C). There was no significant difference in the dry shoot biomass of ‘Goldenwood’ and ‘Vail’ upon P amendment. 

### 3.2. Rhizosphere Microbiome of Legume Crop Varieties under Phosphorus Treatments

The composition of the microbiomes differed according to legume species. There were distinctions in the microbial communities present in common beans, cowpeas, and peas, as evidenced by the significant results obtained from the PERMANOVA analysis (*p* < 0.001) (Figure 2). However, no separation was observed between the fertilized and unfertilized treatments.

### 3.3. Selection of Varieties with Low-P Stress Adaptation

Specific varieties were selected from the different legume crops to determine whether their rhizosphere microbiome might have an impact on P solubilization and acquisition. The varieties were selected for their lack of response to fertilization, potentially indicating an intrinsic ability to obtain P from the soil. In addition, varieties that responded positively to fertilization were selected as controls. 

Among the common bean varieties, Anasazi, ‘Black Beard’, and ‘Cowboy’ were selected. The biomass of Anasazi and ‘Cowboy’ did not demonstrate a noticeable change upon fertilization, while ‘Black Beard’ showed a significant increase (Figure 1A). In the cowpea group, California #5 and Cp4906 were selected. California #5 did not exhibit a significant biomass impact upon fertilization, whereas Cp4906 displayed a notable effect (Figure 1B). In peas, the biomass of ‘Vail’ did not demonstrate a noticeable change upon fertilization, and thus it must have been successful in obtaining P for growth by other means, while the biomass of ‘Melrose’ and ‘Seed Ranch’ showed a significant increase upon P addition (Figure 1C).

### 3.4. P Utilization and Bulk Soil P Concentration

#### 3.4.1. Common Bean (*Phaseolus vulgaris*)

There was a noticeable difference in shoot P concentration and uptake (Figure 3A,B) among the varieties, with Anasazi showing a significantly higher P concentration and uptake as compared to ‘Cowboy’ and ‘Black Beard’. Anasazi also displayed a substantial increase in P availability (Olsen P) in the bulk soil under P-limited conditions, whereas ‘Cowboy’ and ‘Black Beard’ did not show such a difference (Figure 3C).

#### 3.4.2. Cowpeas (*Vigna unguiculata*)

No significant differences in shoot P concentration were observed between the two cowpea varieties (Figure 4A); yet, California #5 had a significantly higher shoot P uptake under the unfertilized treatment compared to Cp4906 (Figure 4B). There were no differences between California #5 and Cp4906 in terms of bulk soil P availability (Figure 4C).

#### 3.4.3. Peas (*Pisum sativum*)

There were significant differences in the shoot P concentrations between the ‘Vail’ and ‘Seed Ranch’ varieties compared to ‘Melrose’ under the unfertilized treatment (Figure 5A). Additionally, ‘Vail’ exhibited greater P uptake in its shoots than ‘Melrose’ (Figure 5B). Furthermore, significant differences were observed in the bulk soil P availability for both the ‘Vail’ and ‘Melrose’ varieties compared to ‘Seed Ranch’ (Figure 5C). 

### 3.5. Differential Abundance Analysis of Bacteria for Selected Unfertilized Varieties

#### 3.5.1. Common Beans (*Phaseolus vulgaris*)

A differential abundance between unfertilized common bean varieties revealed differences in the abundance of distinct bacterial taxa, with certain beneficial microbes (*Mobilisporobacter senegalensis, Sporomusa sphaeroides, Azospirillum brasilense, Anaerospora hongkongensis,* and *Pseudomonas oryzae*) being significantly more abundant (*p* < 0.05) in the Anasazi rhizosphere compared to ‘Cowboy’ and ‘Black Beard’ (Appendix A). 

#### 3.5.2. Cowpeas (*Vigna unguiculata*)

The microbial diversity of cowpea varieties in the rhizosphere soil revealed significant differences in the relative abundance of distinct bacterial taxa under the unfertilized soil treatment. Certain beneficial microbes were found to be significantly more abundant (*p* < 0.05) in both California #5 and Cp4906 (Appendix A). Among those, the following bacteria showed the highest fold change: *Arenimonas daejeonensis, Hyphomicrobium hollandicum Paenibacillus oenotherae,* and *Microlunatus speluncae*. Moreover, in this context, California #5 demonstrated a notable prevalence of beneficial microbes within the rhizosphere soil, underscoring its tendency to harbor a higher abundance of advantageous microorganisms.

#### 3.5.3. Peas (*Pisum sativum*)

Significant differences in the abundance of distinct bacterial taxa were revealed in the analysis of microbial diversity in the rhizosphere soil of various pea varieties under unfertilized conditions. Notably, beneficial microbes were found to be significantly more abundant (*p* < 0.05) in ‘Vail’ compared to ‘Seed Ranch’ and ‘Melrose’ (Appendix A). Among those microbes, the following showed the highest abundance: *Achromobacter xylosoxidan, Rhizobium zeae, Anaerotaenia torta, Cylindrospermum stagnale,* and *Oscillatoria acuminata*. 

### 3.6. Effect of P Fertilization on the Microbiomes of Anasazi, California #5, and Vail

The impact of fertilization on the relative abundance of bacteria expressing P solubilization and mineralization genes in the rhizosphere soil of Anasazi was investigated. A statistically significant difference (*p* < 0.05) in abundance between the fertilized and unfertilized treatments was found (Figure 6A). This disparity in abundance highlighted the impact of P amendment on bacterial populations. Notably, certain bacteria exhibited sensitivity to varying P levels. Among these, Anasazi-cultured bacteria showed a reduction in abundance upon fertilization (Figure 6A). Particularly noteworthy were the following bacteria: *Anaerotaenia torta, Parasegetibacter luojiensis, Pseudomonas sagittaria, Lutispora thermophila, Pseudomonas oleovorans, Pseudomonas resinovorans,* and *Anaerospora hongkongensis* (Appendix A).

In California #5, no significant difference was observed in the relative abundance of the fertilized and unfertilized microbial groups (*p* > 0.05) (Figure 6B). Three specific beneficial taxa, namely *Arenimonas daejeonensis, Hyphomicrobium hollandicum,* and *Paenibacillus oenotherae,* which express P mineralization genes, were identified. These bacteria are described as being insensitive to P fertilization. This insensitivity implies that the microbial community or their activities were not substantially impacted by the addition of P through fertilization. 

The ‘Vail’ variety showed no significant effects of P fertilizer application on the abundance of specific beneficial microbes that expressed P solubilization and mineralization genes, namely *Anaerotaenia torta*, *Cylindrospermum stagnale,* and *Oscillatoria acuminata*. These bacteria are also described as being insensitive to P fertilization (Figure 6C).

## 4. Discussion

### 4.1. The Impact of Phosphorus Solubilization on Microbial Community Composition in P-Insensitive Varieties

Anasazi, a P-efficient common bean, demonstrated enhanced P acquisition when compared to other varieties, as evidenced by Anasazi’s higher shoot P concentration, increased P uptake, and its ability to enhance soil-insoluble P. The rhizosphere of Anasazi exhibited a greater abundance of *Pseudomonas sagittaria* and *Pseudomonas oryzae,* which possess the pyrroloquinoline quinone gene (*pqqC*) *(KEGG: K06137)* associated with P solubility [14,34]. This enhanced P acquisition may be facilitated not only by the identified bacterial species but also by the root exudates that shape the microbial community in the rhizosphere. Root exudates, which include a variety of organic acids and other compounds, can modify the soil environment to favor the growth of beneficial microbes such as PSB and arbuscular mycorrhizal fungi (AMF), thereby improving P bioavailability [35,36]. Moreover, the presence of *Pseudomonas resinovorans*—a bacterial species harboring both the *pqqC* gene and the alkaline phosphomonoesterase gene (*phoD*) (KEGG: K01113)—further contributed to the mineralization of organic P. Anasazi also had an abundance of other P mineralizing bacteria with the presence of the phosphatase *phoA* gene (KEGG: K01077) and the phosphatase *phoD* gene, such as *Mobilisporobacter senegalensis, Sporomusa sphaeroides, Anaerotaenia torta, Anaerospora hongkongensis, Asticcacaulis endophyticus Herbinix luporum,* and *Parasegetibacter luojiensis*. These bacteria contain genes that encode alkaline phosphomonoesterase, which plays a significant role in the mineralization of organic P [37,38]. These findings suggest that the Anasazi varieties may establish associations with these bacteria to solubilize reserves of legacy P, thus highlighting its efficient prioritization of P acquisition strategies. Notably, these bacteria exhibit growth in response to increased alkaline phosphatase activity in the rhizosphere [39], further emphasizing the efficient utilization of P acquisition strategies. Moreover, some of these bacteria enriched in the Anasazi rhizosphere (*Pseudomonas sagittaria, Sporomusa sphaeroides, Pseudomonas oryzae Herbinix luporum, Anaerospora hongkongensis*, and *Anaerotaenia torta*) exhibit dual gene expression; they not only possess genes responsible for the solubilization and mineralization of P but also contain the genes *nifH* (encoding nitrogenase Fe protein) and *nifD* (encoding nitrogenase Mo-Fe protein), which are involved in N fixation. This evidence suggests that Anasazi may cultivate relationships with these bacteria. 

Compared to other cowpea varieties, specific bacteria were enriched in the California #5 rhizosphere including *Paenibacillus oenotherae* (contains *phoA*), *Microlunatus speluncae* (contains *phoA*)*, Hyphomicrobium hollandicum* (contains *pqqC*), and *Arenimonas daejeonensis* (contains *PHO* (encodes acid phosphatase, KEGG: K01078)). The presence of these bacteria in the California #5 rhizosphere likely contributes to a range of advantages, such as improved acquisition of P, higher P concentration in the shoots, and the ability to access legacy P (Figure 3B). In a related study, microorganisms possessing *phoA* and *phoD* genes played a crucial role in obtaining P under low-P conditions [40]. The authors observed that—as rice plants matured—labile inorganic P increased with the activity of bacteria that express genes for solubilizing rhizosphere P. This finding further supports the enhanced P-efficiency of the California #5 cowpea variety, which may result from the release of bound phosphate anions from soil minerals and the enzymatic hydrolysis of phosphor-esters facilitated by phosphatase enzymes [41]. 

The ‘Vail’ pea variety displayed an increase in shoot P concentration compared to other varieties, leading to enhanced P uptake in the shoots and facilitating access to otherwise insoluble phosphates in the soil (Figure 3C). Beneficial microbes expressing the *pqqC* gene were more abundant in the ‘Vail’ rhizosphere. One bacterium present, containing the *pqqC* gene, was *Anaerotaenia torta*. Additionally, the rhizosphere of ‘Vail’ was enriched with *Cyanothece sp. PCC 7425, Oscillatoria nigro-viridis, Oscillatoria acuminata*, *Shinella sp. HZN7, Microcoleus sp. PCC 7113, Rhizobium zeae, Paenibacillus agaridevorans,* and *Devosia riboflavina*. These bacteria have been found to express the gene for both acid phosphatase (*PHO*) and alkaline phosphatase (*phoA, phoD*), potentially causing increased availability of bioavailable P in the rhizosphere of the ‘Vail’ variety [40]. Additionally, ‘Vail’ may establish associations with N-fixing bacteria, such as *Rhizobium zeae*. Although these relationships primarily benefit N acquisition, they can indirectly impact phosphorus availability as well [42]. 

### 4.2. Impact of the Phosphorus Application on Beneficial Phosphorus-Solubilizing Bacteria in P-Insensitive Varieties of Anasazi, California #5, and Vail

One of the main goals of this study was to identify novel beneficial bacteria that could solubilize legacy P to provide additional P to crops. However, these bacteria would need to be applied in conjunction with decreased levels of P fertilizer to provide a P boost that could maintain yields. This is of particular importance when growers eventually deal with future P fertilizer shortages while attempting to maintain viable agricultural practices. Thus, a key aspect of these microbial P solubilizers would be their ability to be unaffected by P fertilization (i.e., insensitive to P fertilization). To address this, we compared the fertilized and unfertilized treatments in P-insensitive varieties (Anasazi, California #5, and ‘Vail’) described above.

Phosphorus solubilizing bacteria in Anasazi, including *Mobilisporobacter senegalensis, Sporomusa sphaeroides, and Azospirillum brasilense*, displayed sensitivity to P levels. When P was applied, these bacteria experienced a decrease in their relative abundance, indicating a negative impact on their population dynamics. These findings align with a study conducted where the presence of P fertilizer in soils led to a notable decrease in the relative abundance of proteobacteria members, specifically the Gammaproteobacteria class, Pseudomonadales order, and *Acinetobacter* genus [43]. In contrast, the relative abundance of P solubilizing and mineralizing bacteria in California #5 and ‘Vail’ varieties did not decrease when P fertilizer was applied. In California #5, there was no significant difference in the relative abundance of microbial groups between the fertilized and unfertilized samples, indicating minimal impact of P fertilizer addition on the microbial community. Four P mineralizing bacteria *(Arenimonas daejeonensis, Hyphomicrobium hollandicum, Paenibacillus oenotherae,* and *Microlunatus speluncae*) were among those species that did not change in abundance, suggesting their ability to function effectively may have been unchanged with fertilization. Similarly, in the ‘Vail’ variety, the addition of P fertilizer did not significantly affect the abundance of specific beneficial microbes harboring P solubilization and mineralization genes, such as *Cylindrospermum stagnale* and *Anaerotaenia torta*. This implies that these bacteria maintained some resiliency when introduced to P fertilization. The insensitive response of these bacteria to P fertilization aligns with the recent discovery of a phosphate-insensitive phosphatase gene (*PafA*) in Bacteroidetes, emphasizing the efficient utilization of organophosphorus substrates by plant-associated *Flavobacterium* species [44]. The resilience of these P-insensitive bacteria in the presence of added P fertilizers suggests a potential for these microbes to function in various soil P conditions. This ability may be supported by root exudates that not only serve as nutrients but also as signaling molecules, influencing the composition and function of the rhizosphere microbiome [36].

## 5. Conclusions

Upon review of the existing body of work, it was evident that P-insensitive microbes (e.g., *Arenimonas daejeonensis, Hyphomicrobium hollandicum, Paenibacillus oenotherae,* and *Microlunatus speluncae)* offer novel opportunities for promoting sustainable agricultural practices and improving nutrient utilization. Identifying specific root exudates or other plant attributes that support the growth of these microbes, while tailoring them to different plant varieties, could be integrated into breeding initiatives. This approach holds promise for enhancing agricultural practices in an environmentally friendly manner, contributing to the long-term sustainability of our food systems. Furthermore, our research underscores the significance of legumes as model systems for developing technologies aimed at enhancing P sustainability. 

Building upon these insights, the findings of this study highlight the significant impact that microbes have on the rhizosphere environment. Our research demonstrates that the presence and activity of specific microbial communities are integral to the modulation of P bioavailability in the soil. These microbial interactions within the rhizosphere can lead to increased nutrient availability and better plant health, ultimately influencing crop yield and sustainability. Such knowledge is crucial for the development of next-generation agricultural practices that leverage the natural soil microbiome for enhanced productivity and ecological balance. 

## Figures and Tables

**Figure 1 microorganisms-12-00353-f001:**
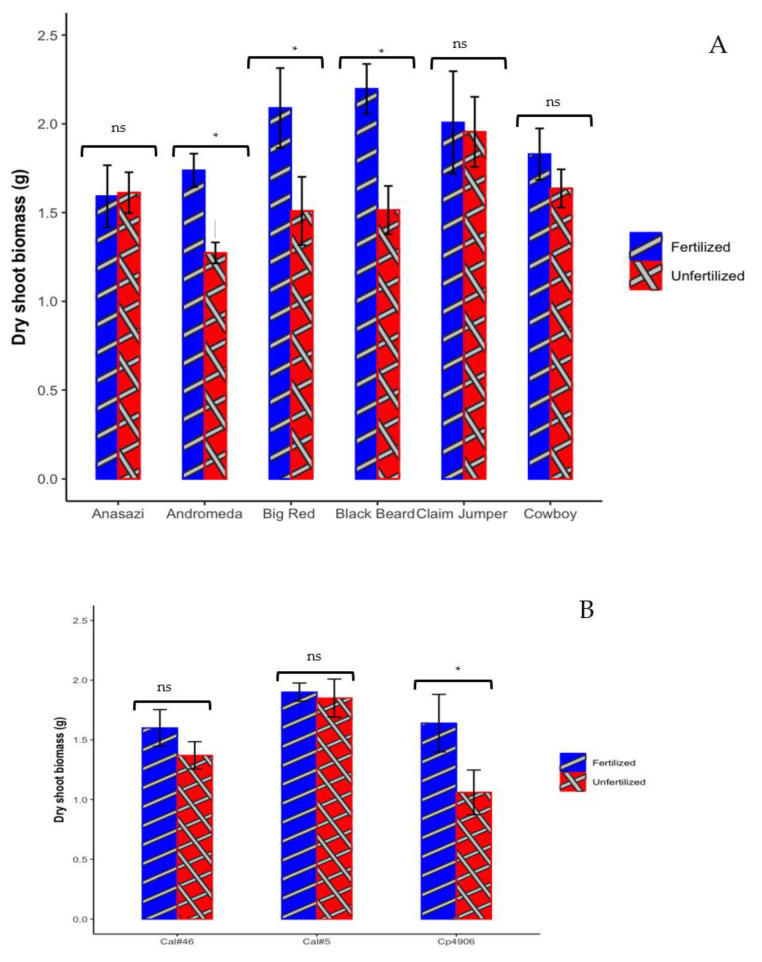
Dry shoot biomass presented as mean ± SEM for (**A**) six varieties of common beans: Anasazi, ‘Andromeda’ (Great northern), ‘Big Red’ (light red kidney beans), ‘Black Beard’ (black), ‘Claim Jumper’ (Mayacoba), and ‘Cowboy’ (pinto), (**B**) cowpeas: California #46 (Cal#46), California #5 (Cal#5), and Cp4906 (landrace from Portugal), and (**C**) peas: ‘Goldenwood’ (yellow), ‘Koyote’ (yellow), ‘Melrose’ (Austrian winter pea), ‘Seed Ranch’ (Austrian winter pea), and ‘Vail’ (green). A two-way ANOVA was used to determine significant differences between P treatments. * represents a significant difference between treatments at a *p* < 0.05. ns = not significant. Error bars represent the standard error of the mean.

**Figure 2 microorganisms-12-00353-f002:**
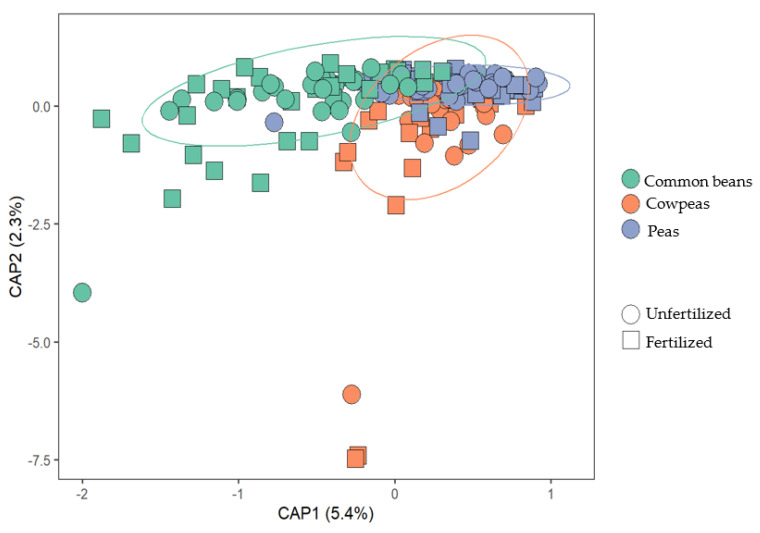
Distance-based redundancy analysis (db-RDA) of the soil microbiome sequencing data from common beans, cowpeas, and peas under two different P fertilizer treatments. The colors on the graph indicate green for common beans, orange for cowpeas, and blue for peas. The shapes represent circles for unfertilized and squares for fertilized.

**Figure 3 microorganisms-12-00353-f003:**
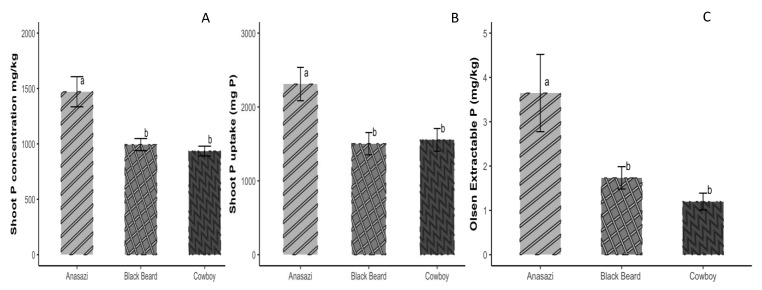
Anasazi, Black Beard, and Cowboy common beans. (**A**) Shoot P concentration, (**B**) shoot P uptake (mg P), and (**C**) bulk soil Olsen (available) P concentration without P fertilization. One-way ANOVA was used to assess differences among the varieties. Different letters above error bars indicate a significant difference at *p* < 0.05. Error bars represent the standard error of the mean.

**Figure 4 microorganisms-12-00353-f004:**
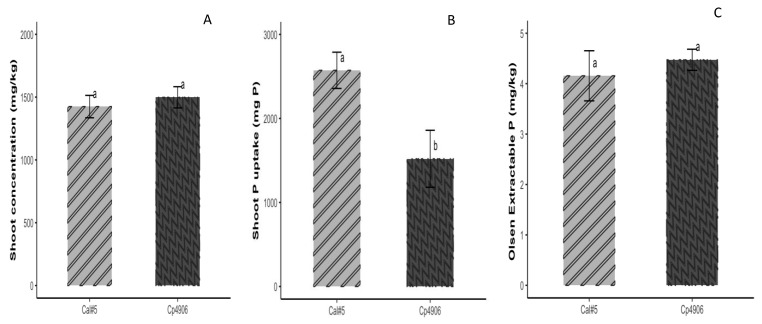
Cowpea varieties California #5 (Cal#5) and Cp4906. (**A**) Shoot P concentration, (**B**) shoot P uptake (mg P), and (**C**) bulk soil Olsen (available) P concentration without P fertilization. One-way ANOVA was used to assess differences between the varieties. Different letters above error bars indicate a significant difference at *p* < 0.05. Error bars represent the standard error of the mean.

**Figure 5 microorganisms-12-00353-f005:**
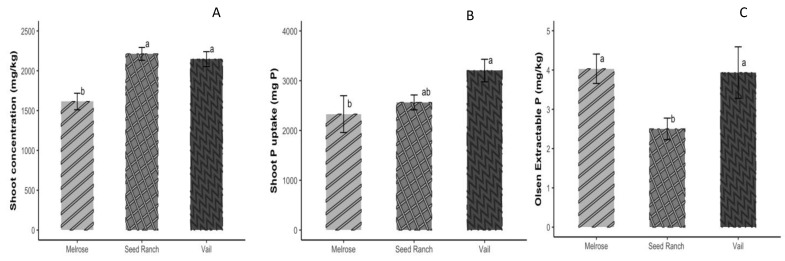
Pea varieties Seed Ranch, Melrose, and Vail. (**A**) Shoot P concentration, (**B**) shoot P uptake (mg P), and (**C**) bulk soil Olsen (available) P concentration without P fertilization. One-way ANOVA was used to assess differences between the varieties. Different letters above error bars indicate a significant difference at *p* < 0.05. Error bars represent the standard error of the mean.

**Figure 6 microorganisms-12-00353-f006:**
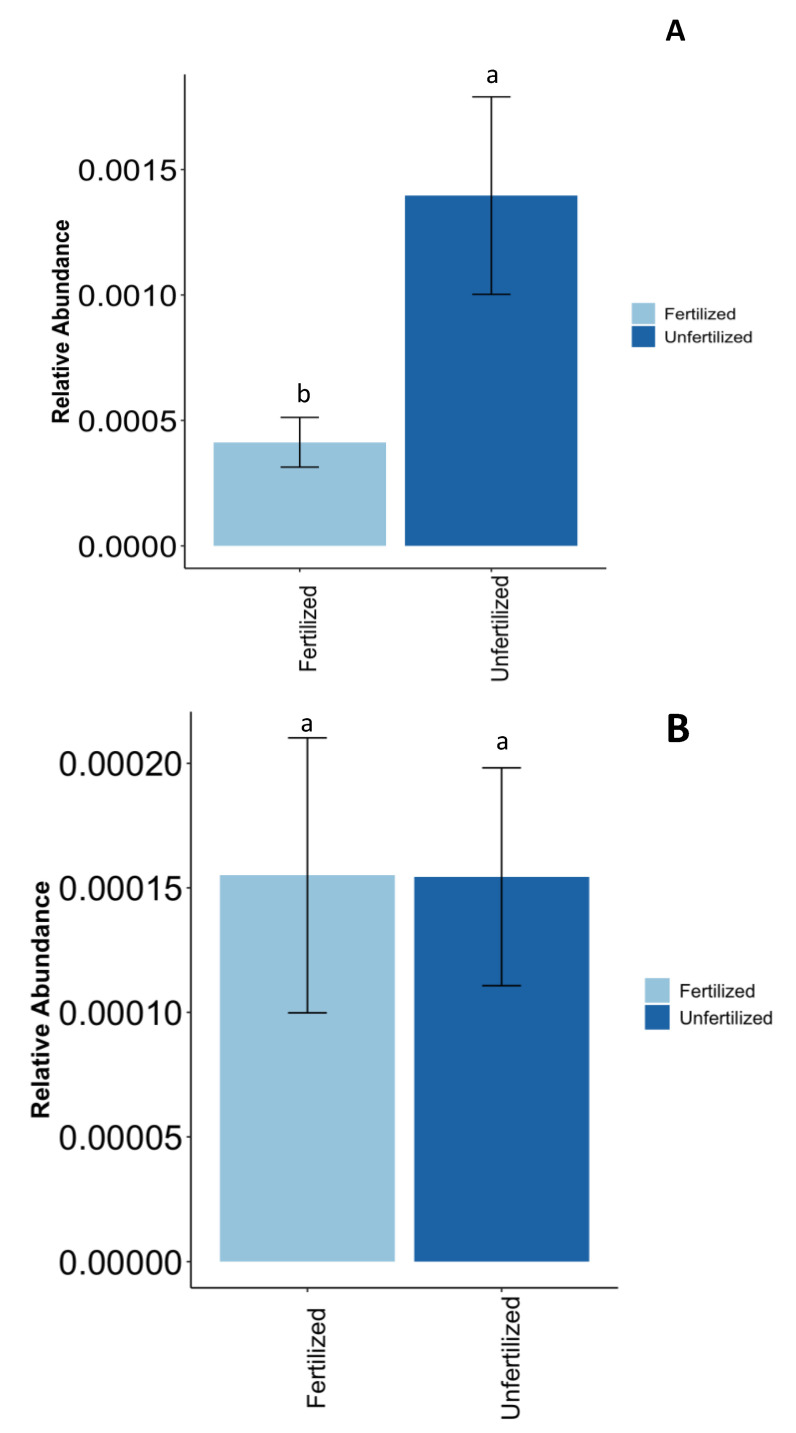
The relative abundance of bacteria-expressing genes related to P solubilization and mineralization in the rhizosphere under fertilized and unfertilized soil for (**A**) Anasazi (common bean), (**B**) California #5 (cowpea), and (**C**) Vail (pea). Different letters above bars represent significant differences at *p* < 0.05. Error bars represent the standard error of the mean.

**Table 1 microorganisms-12-00353-t001:** Analysis of the physical and chemical properties of the soil mix used in the study. The soil mix consisted of one part sand and one part soil.

Parameter	Value
pH	8.4
Soluble salts (dS m^−1^)	0.47
Organic Matter LOI, %	0.9
Nitrate-N (mg kg^−1^)	3.9
Total P (mg kg^−1^)	371
Olsen P (mg kg^−1^)	2.6
Water holding capacity (%)	12.0

## Data Availability

The data that support the findings of this study are available from the corresponding author upon reasonable request.

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
