# Peer review of "Harnessing Phosphorous (P) Fertilizer-Insensitive Bacteria to Enhance Rhizosphere P Bioavailability in Legumes"

_microorganisms, 2024, doi:10.3390/microorganisms12020353_

Round 1

Reviewer 1 Report

Comments and Suggestions for Authors

This publication presents interesting results related to the identification of some P fertilizer-insensitive bacteria in the rhizosphere of legumes and their potential impact on soil P availability and its uptake. The adopted approach in this study was interesting since it could provide more insights into the use of these bacteria to promote sustainable agriculture and to reduce human dependency on a non-renewable P resource.

The manuscript was well introduced, and the authors adopted adequate methods with discussion of the different obtained results. However, the manuscript needs major revisions to be suitable for publication in Microorganims.

General comments

- Comment 1: The English of this manuscript needs some improvements in addition to punctuation and typos issues

- Comment 2: Some details are missing in M&M section.

- Comment 3: The discussion section should be more developed.

Other comments

- Abstract

L28: please italicize the genera scientific name. Please check throughout the manuscript.

Keywords: please avoid the use of abbreviation.

Introduction:

Please include a clear hypothesis at the end of the introduction section.

- M&M

L115: “Pots (110 x 110 x 130 cm)”, do you mean “Pots (110 x 110 x 130 mm)”?

L120: “with 0.098 g. TSP (0.045 g P2O5)”, it seems that you applied the same amount of fertilizer for the three species of legumes while there basically have different needs regarding fertilization rates. Why?

L123; Please add some details on seeds pretreatment.

L123-125: please specify the number of treatments and repetitions.

L126-127: “The 126

soil consisted of one part sand and one part soil.”, please place in the text.

L129-130: Please specify other growth conditions such as relative humidity and light intensity.

L144-145: For ICP-OES, please add details on the manufacturer and the country. Please check throughout the manuscript for other equipment.

L148: please add a space between the value and the unit.

L150: please remove the additional space and check all other additional or missing ones in the whole manuscript.

- Results

Please use some percentage values in this section.

Figure 1A: Please correct the offset between significance bars and histograms.

L243: Please add the “Distance-based redundancy analysis” in the statistical analysis in M&M section.

Please change “3.5., 3.6 and 3.7 subsections” to “3.5. P utilization and Bulk Soil P and 3.6. Differential Abundance Analysis of Bacteria for Unfertilized Varieties” and presents the three species results inside these two subsections and edit accordingly.

- Discussion

Please provide a more detailed discussion of your results by including the role of root exudates to explain the obtained results and underlying that the obtained effects are not exclusively related to PSB but also to other beneficials microorganisms including AMF since there are also known to alter P bioavailability.

Please add more literature references.

L395 and L431: please correct the citation form and check throughout the manuscript.

Comments on the Quality of English Language

The English of the manuscript needs moderate editing.

Reviewer 2 Report

Comments and Suggestions for Authors

This is a research report in which the authors propose P-insensitive microbes to optimize P utilization and drive future sustainable agricultural practices to reduce human dependence on a non-renewable resource. Important research since phosphorus is a non-renewable element. The investigation seems to be well designed and has a logical sequence, however, its presentation is poor, it seems like a first draft of the report. On the other hand, there are many errors in the format, and above all, the Conclusions section is missing. It is recommended that the authors carry out an in-depth examination of the manuscript.

It is recommended that the authors highlight the findings of this research and include the effect of microbes on the rhizosphere.

More comments on the manuscript.

Comments on the Quality of English Language

Comments on the manuscript.

Author Response

Dear Reviewer, # 2

We are sincerely grateful for your comprehensive review and insightful comments. Please find our responses to your points below, which we believe have further refined our manuscript:

Point 1: Improvement of Manuscript Presentation:

Response 1: The manuscript has undergone a thorough revision to ensure a coherent structure and flow that aligns with the high standards of publication.

Point 2: Inclusion of Conclusions Section:

Response 2: A detailed conclusions section has been crafted, summarizing the pivotal discoveries of our research and their broader implications (line 468)

Point 3: Emphasis on Microbial Effects in the Rhizosphere:

Response 3: The suggested highlights regarding the impact of microbes on the rhizosphere have been added (see Lines 478-485).

We trust that these revisions adequately address your concerns and significantly elevate the manuscript. We are indebted to you for your valuable contributions that have undeniably enhanced our work.

With warm regards,

Round 2

Reviewer 1 Report

Comments and Suggestions for Authors

The authors satisfied all the raised comments. I recommand the publication of the manuscript in the current form.